# Targeting *RB1* Loss in Cancers

**DOI:** 10.3390/cancers13153737

**Published:** 2021-07-25

**Authors:** Paing Linn, Susumu Kohno, Jindan Sheng, Nilakshi Kulathunga, Hai Yu, Zhiheng Zhang, Dominic Voon, Yoshihiro Watanabe, Chiaki Takahashi

**Affiliations:** 1Division of Oncology and Molecular Biology, Cancer Research Institute, Kanazawa University, Kanazawa 920-1192, Japan; dr.painglinn@gmail.com (P.L.); skohno@staff.kanazawa-u.ac.jp (S.K.); jindan000000@gmail.com (J.S.); nilakshi.kulathunga@gmail.com (N.K.); adam9426@hotmail.com (H.Y.); zhangzhiheng930713@yahoo.co.jp (Z.Z.); 2Yangon General Hospital, Yangon, Myanmar; 3Institute of Frontier Sciences Initiative, Kanazawa University, Kanazawa 920-1192, Japan; dvoon@staff.kanazawa-u.ac.jp; 4iCREK, Kanazawa University Hospital, Kanazawa 920-8641, Japan; yoshi.watanabe@staff.kanazawa-u.ac.jp

**Keywords:** RB1, E2F, chromatin instability, synthetic lethality, collateral lethality

## Abstract

**Simple Summary:**

Irreversible defects in RB1 tumor suppressor functions often predict poor outcomes in cancer patients. However, the *RB1*-defecient status can be a benefit as well for them, as it generates a variety of vulnerabilities induced through the upregulation of RB1 targets, relief from functional restrictions due to RB1 binding, presence of genes whose inactivation cause synthetic lethality with *RB1* loss, or collateral synthetic lethality owing to simultaneous loss of neighboring genes.

**Abstract:**

Retinoblastoma protein 1 (RB1) is encoded by a tumor suppressor gene that was discovered more than 30 years ago. Almost all mitogenic signals promote cell cycle progression by braking on the function of RB1 protein through mono- and subsequent hyper-phosphorylation mediated by cyclin-CDK complexes. The loss of RB1 function drives tumorigenesis in limited types of malignancies including retinoblastoma and small cell lung cancer. In a majority of human cancers, RB1 function is suppressed during tumor progression through various mechanisms. The latter gives rise to the acquisition of various phenotypes that confer malignant progression. The RB1-targeted molecules involved in such phenotypic changes are good quarries for cancer therapy. Indeed, a variety of novel therapies have been proposed to target *RB1* loss. In particular, the inhibition of a number of mitotic kinases appeared to be synthetic lethal with RB1 deficiency. A recent study focusing on a neighboring gene that is often collaterally deleted together with *RB1* revealed a pharmacologically targetable vulnerability in *RB1*-deficient cancers. Here we summarize current understanding on possible therapeutic approaches targeting functional or genomic aberration of RB1 in cancers.

## 1. Introduction

The discovery of *RB1* gene had paved an avenue toward understanding the complicated functions of the human genome in preventing cells from carcinogenesis [1,2]. *RB1* gene was thought to be mechanistically involved in the initiation of retinoblastoma as its inactivation following genetic mutation or deletion was prevalently identified in the pedigrees of hereditary cases [3]. This notion was later experimentally proven by the induction of retinoblastoma in mice following the ablation of multiple *Rb* family members [4]. Together with the presence of other cancer driving mutations in genes such as *Trp53*, *RB1* loss can provoke carcinogenesis from various types of tissue. This suggests that RB1 may in fact be involved in the initiation and/or progression of previously unexpected varieties of cancer [5]. Of note, RB1 inactivation is detected in close to 90% of sporadic small cell lung cancers (SCLC), suggesting a contribution to their initiation, while its germ line mutation also often predisposes to SCLC [6]. Reportedly, RB1 inactivation occurs in 20~40% of osteosarcoma cases, yet it is thought to contribute to tumor progression based on pathological examination and linkage with unfavorable outcome [7,8]. RB1 reconstitution in *RB1*-deficient osteosarcoma-derived cells does not always completely attenuate their malignant phenotypes, presumably due to coexisting functional aberration in *Trp53* tumor suppressor or other driver mutations [9,10]. These findings also indicate that RB1 may exert varied functions depending on cancer tissue type and timing during the course of tumor development.

Similar to osteosarcoma, RB1 is suggested to be inactivated during their progression in a majority of cancer types [7]. One of the best examples is prostate cancer. In primary/non-metastatic prostate cancers, *RB1* deletion is identified with less than 10% prevalence, but this rises to higher than 30% in those attaining metastatic and/or castration-resistant phenotypes [11]. In addition to prostate cancer, many common types of cancers, including non-small cell lung cancer (NSCLC), breast cancer, bladder cancer, chronic myelogenous leukemia (CML), high-grade glioma, esophageal cancer, head and neck cancer and sarcoma, are thought to gain *RB1* deficiency during their malignant progression [7].

The roles of RB1 in suppressing malignant progression of prostate cancer may not be solely explained by its primary function to constrain cell cycle progression. For example, RB1 controls the transcription of androgen receptor (AR) through the E2F family of transcription factors, hence its inactivation could enhance the refractoriness to endocrine therapy [12]. Moreover, the depletion of RB1 from *RB1*-proficient prostate cancer cells promoted gain of stem cell-like properties through increased expression of interleukin-6 (IL-6) and lysyl oxidase (LOX) [13]. Similarly, RB1 inactivation in breast cancers enhances stem cell-like behaviors and malignant progression through IL-6 and following activation of signal transducer and activator of transcription 3 (STAT3) activation [14,15,16,17,18]. *RB1* loss in breast cancer cells activates fatty acid oxidation (FAO) and induces following Jun kinase (JNK) activation. Activated JNK then stimulates IL-6 production leading to gain of undifferentiated characters which are mediated by cell-autonomous STAT3 activation [14]. Later, the enhanced FAO following *RB1* loss was explained by AMP-activated protein kinase (AMPK) activation and subsequent phosphorylation (downregulation) of acetyl-CoA carboxylase (ACC) [16]. Accordingly, the blockade of IL-6 by neutralizing antibody suppressed breast cancer progression is induced following RB1 inactivation [14]. Loss of *Trp53* in mice generates well differentiated soft tissue sarcoma with low prevalence; additional loss of *Rb1* converts this to undifferentiated pleomorphic type [17]. An increase in lineage plasticity has been observed in mouse *Trp53*-null osteosarcoma cells following *Rb1* deletion [19]. Identification of molecules mediating RB1 functions in controlling undifferentiated characters of tumor cells may endow us new tools to target cancer stem cells [18]. Taken together, in addition to targeting aberrant cell cycle progression, functional suppression of the proteins aberrantly expressed as a result of RB1 inactivation may also be beneficial in cancer treatment.

This review aims to firstly shed light on such gene products and then focus on genes whose inactivation might exhibit synthetic lethality when RB1 is simultaneously inactivated. Finally, this review will discuss putative approaches to treat *RB1*-deficient cancers by targeting comparatively large-scale chromosomal aberration involving the *RB1* loci.

## 2. Targeting Gene Products Upregulated Following RB1 Inactivation

The RB1/E2Fs/DPs complex primarily suppresses target gene transcription by recruiting histone deacetylase (HDAC). Loss of RB1 function liberates its most important counterpart, E2Fs, and these primarily transactivate target genes by recruiting histone acetylase (HAT). The target genes of the “transactivating” E2Fs (E2F1, E2F2 and E2F3a) contain a number of genes enrolled in cell cycle progression, DNA synthesis and replication, DNA damage repair and apoptosis [20]. In line with this, the deletion of *E2f1* suppressed pituitary and thyroid tumorigenesis and prolonged the life span in *Rb1*-heterozygous mice [21]. Likewise, the loss of *E2f2* or *E2f3* in MMTV-Myc transgenic mice resulted in the cessation of mammary carcinogenesis [22]. E2F1 is frequently overexpressed in metastatic melanoma, and inhibition of E2F1 induced apoptosis and cellular senescence [23]. In specific types of cancer, higher expression of some of E2F family members correlates with advanced stage and poor prognosis [24]. Therefore, targeting E2Fs may yield certain benefits in cancer therapy. The E2F inhibitor HML006474 was designed based on a virtual screening of compounds that possibly inhibit DNA binding of the E2F4/DP2 complex. The E2F4-null cells were less sensitive to this compound compared to wild type, which supports the specificity of its action [25]. However, to date, no small molecules specifically targeting any of transactivating E2Fs have been described nor tested in clinical trials.

As E2Fs target a wide variety of genes that could promote or suppress tumor development, it is much safer to consider strategies to target individual E2F target. As numerous review articles have already expounded on this point, this article picks up only a few. In particular, although cyclin E and cyclin dependent kinase 2 (CDK2) are representative targets of E2Fs during G1/S, and inhibitors to CDK2 are undergoing clinical trials [26], the efficacy of these agents and their dependency on the presence of intact RB1 remains a scantly addressed topic.

E2Fs target numerous genes involved in replication control, nucleotide synthesis, DNA damage response/repair, chromatin structure and mitotic progression. RB1 inactivation induces chromosomal instability (CIN), at least partially, by upregulating spindle assembly checkpoint protein mitotic arrest deficient 2 (MAD2). Overexpressed MAD2 may contribute to induced aneuploidy and stabilization of erroneous attachment of mitotic spindles [27,28]. Dysregulated expression of MAD2 is correlated to cancer progression and poor prognosis [29]. A small molecule, M2I-1, targeting MAD2-CDC20 interaction has been developed [30]. A report suggested that M2I-1 sensitizes cancer cells to anti-mitotic reagents [31] but, thus far, no report has tested its efficacy in *RB1*-deficient context. Many of mitotic kinases are upregulated following E2F dysregulation and inhibition of some of them has been indicated to be synthetically lethal with RB1 inactivation (see below).

Besides MAD2, numerous possibly druggable molecules have been indicated to be involved in the *RB1* loss-induced CIN [32]. Aurora A and polo-like kinase 1 (PLK1) are upregulated following RB1 depletion, which may lead to aneuploidy or other mitotic abnormalities, further causing malignant progression [33]. Synthetic inhibitors to these kinases are now in clinical trials (Table 1). Mis-localization of CAPD3/condensin II has been proposed to mediate centromere dysfunction induced by *RB1* loss [34]. However, synthetic inhibitors to this molecule are under development and it is not clear whether such inhibitors could exhibit anti-tumor activity.

In addition to mitotic catastrophe, replicative catastrophe induced by abnormally enhanced cell cycle progression induced by gain of function by E2Fs could provide a certain vulnerability especially upon chemotherapy or radiation therapy [35]. The observation that RB1 inactivation in thyroid calcitonin-producing (C) cells or fibroblastic cells induced DNA damage response followed by senescence supports this speculation [36].

As alluded to the introduction, mostly in an E2F-dependent manner, *RB1* status affects the milieu surrounding tumor cells including chemokine/cytokine secretion, extracellular matrix, immune cells or remote organs to be metastasized. The molecules mediating such functions of RB1 could serve as good therapeutic targets [37,38]. C-C motif chemokine 2 (CCL2) appeared to be secreted from *RB1*-deficient breast cancer cells, thereby such tumor cells recruit immunosuppressive cells including myeloid-derived suppressive cells (MDSC), regulatory T cells (Treg) or macrophages, allowing malignant progression following RB1 inactivation. Moreover, blockade of CCL2 signaling in host mice by genetically deleting *CCR2*, encoding the receptor for CCL2, almost completely abrogated mammary carcinogenesis from *MMTV-cre*; *Rb1*^flox/flox^ mice [14,38]. RB1 inactivation has been linked to upregulation of PD-L1 expression through direct interaction with NF-κB protein p65. Surprisingly, a particular phosphorylated form of RB1 inactivates p65 and therefore attenuates expression of PD-L1 and cIAPs. The peptide mimicking this phosphorylated form of RB1 improved the therapeutic efficacy in radiotherapy by suppressing radiation-induced PD-L1 expression [39]. These findings make clear contrast with the enhanced immunogenicity induced by locking RB1 in unphosphorylated form by the treatment with CDK4/6 inhibitor [40], and suggest that *RB1* status might affect the efficacy of therapy by immune checkpoint inhibitors (ICIs).

A while ago, an unexpected twist arose from *C. elegans* studies that indicated lin-35, an RB1 orthologue, might be placed upstream of Ras signaling during vulval development [41,42]. In cultured cells, including embryonic fibroblasts and thyroid C cells, K-Ras and N-Ras are activated 5~10 times greater following *RB1* loss, and this may promote or suppress tumor development in a highly context-dependent manner [36,43,44,45,46,47]. It has been estimated that mutation of K-Ras or N-Ras in the context of *RB1* loss exerts a mitogenic effect that is 50~60 times greater than that of corresponding wild type Ras in the presence of RB1. Mechanistically, the RB1-Ras nexus has been explained by farnesylation and geranylgeranylation to anchor cytosolic Ras proteins to lipid bilayers, which are innervated by E2Fs and SREBPs [36,48], thus this axis is pharmacologically targetable. These findings led researchers to address RB1 functions in cholesterol and then lipid metabolism. ELOVL fatty acid elongase 6 (ELOVL6) and fatty acid desaturase 1 (SCD1) have been linked to RB1 via E2Fs and SREBPs, which explained the profound impact of *RB1* loss on the lipidomic landscape [49]. The oleic acid, one of the immediate products of ELOVL6 and SCD1, appeared to promote mammary carcinogenesis by stabilizing c-Myc through GPR40 which is a receptor for oleic acid, providing a good rationale to develop antagonists to this receptor to prevent breast cancer associated with high fat diet in puberty [50].

Although the dependence on E2Fs is still unclear, human retinoblastoma cells highly express a proto-oncogene spleen tyrosine kinase (SYK) and depend on its expression for survival. Treatment of genetic retinoblastoma model with synthetic SYK inhibitor resulted in dramatic suppression of disease progression and prolongation of survival [51]. Furthermore, in retinoblastoma cells, ubiquitin-like, containing PFD and RING finger domains 1 (UHRF1) and helicase, lymphoid specific (HELLS) are overexpressed, which is linked to epigenetic activation of SYK [52,53].

## 3. Partners Other Than E2Fs Liberated upon *RB1* Loss

More than 300 proteins have been indicated to directly bind to RB1 [48]. Varied mono-phosphorylation status of RB1 (possibly 14 variations) may increase the complexity of the relationship with specific binding partners [54]. As RB1 is typically not abundantly expressed in tissues, RB1 may carefully select partners depending on the cellular context. A number of chromatin modifiers, cyclins and signaling molecules, such as apoptosis signal-regulating kinase 1 (ASK1) equipped with LxCxE motif, bind to RB1 at the site also targeted by a number of oncogenic virus products carrying LxCxE motif including adenovirus E1A, simian virus 40 large T antigen and human papilloma virus E7 [55]. Such chromatin modifiers include attractive targets of cancer therapy. Notably, 5-azacytidine targets DNMT1 and has been used for the treatment of myelodysplastic syndrome [56]. Similarly, the utility of pan-HDAC inhibitors has long been evaluated in a variety of cancers [57]. The Suv39H1 inhibitor chaetocin exhibited significant therapeutic efficacy against acute myeloid leukemia when combined with other epigenetic drugs that include suberoylanilide hydroxamic acid (HDAC inhibitor) and JQ-1 (bromodomain inhibitor) [58]. However, how the loss of *RB1* impacts the efficacies and sensitivities of these molecules is not well studied. One notable exception is TAI-1, which targets Hec-1 and exhibits synergy with multiple chemotherapeutic agents in the treatment of leukemia, breast and liver cancer. In these settings, the status of *RB1* and *Trp53* influenced the sensitivity to TAI-1 [59].

SKP2 directly binds to RB1 [60], whereupon *RB1* loss liberates SKP2 to promote the proteasomal degradation of p27^KIP^. SKP2 was initially identified as a cyclin A binding protein and later found to serve as a component of the SCF^SKP2^ E3 ubiquitin ligase complex targeting various substrates containing many of cell cycle regulators, including p27^KIP^. Additional loss of *Skp2* loci appeared to increase the degree of apoptosis in *Rb1*-nullizygous mouse embryonic fibroblasts (MEFs) in an E2F-dependent manner. Similarly, *Skp2* loss increased apoptotic cells in *Rb1*-deficient pituitary tumor developed in mice, also in an E2F-dependent manner [61]. Loss of *SKP2* stabilizes p27^KIP^ and simultaneously exposes cyclin A to p27^KIP^. Additionally, the loss of *SKP2* lessens cyclin A activity to attenuate E2F1 function. Consequently, overactivated E2F1 behaves more proapoptotic rather than proliferative in the absence of *RB1* [61].

Deficiency in terminal differentiation following RB1 inactivation would be another possible target in cancer treatment. Many of the tissue-specific transcription factors, including MYOD, CCAAT enhancer binding protein (C/EBP), glucocorticoid receptor (GR), GATA-binding factor 1 (GATA-1), PU-1, core-binding factor alpha 1 (CBFA-1), pancreas duodenum homeobox 1 (PDX1), runt-related transcription factor 2 (RUNX2) and nuclear factor of IL-6 expression (NF-IL6), directly or indirectly collaborate with RB1 to determine lineage specificity and to induce terminal differentiation [62]. As such, the loss of RB1 promotes undifferentiated behaviors of cancer cells, though the extent to which tissue-specific transcription factors also contribute to this phenotype is still unclear. Added to this, the functional recovery of tissue-specific transcription factors by pharmacological approaches may be difficult.

Yet how the loss of *RB1* promotes undifferentiated behaviors of cancer cells might be at least partially explained by the pivotal roles of RB1 in controlling pluripotency factors including octamer-binding transcription factor 4 (OCT4) and SRY-box 2 (SOX2), by directly binding to the regulatory elements for these genes [63]. This study has been done in the context of inducible pluripotent stem cells (iPSs). Additionally, as abovementioned, IL-6 and LOX have been proposed to explain the effect of *RB1* loss on the malignant progression of prostate and breast cancer [13,14]. Interference to these machineries may enable cancer therapy by inducing spontaneous differentiation.

## 4. Synthetic Lethality with RB1 Inactivation

It is commonly held that loss-of-function mutations or deletions of tumor suppressor genes are hard to pharmacologically target in cancer therapy. One possible approach to target cancers with an inactivated tumor suppressor is to find synthetic lethal targets in such tumor cells [64]. The discovery of synthetic lethal interaction between breast cancer susceptibility gene 1/2 (BRCA1/2) tumor suppressors and poly ADP-ribose polymerase (PARP) DNA repair protein provided an opportunity to use PARP inhibitors in clinical cancer therapy [65]. With the aim of identifying genes that are in synthetic lethal relationship with *RB1*, RNAi, shRNA, CRISPR/Cas9i or chemical libraries have been often employed on high-throughput screening bases. *RB1* mutation contributes to the development of SCLC (see above). However, it has been difficult to develop methods to directly target *RB1* mutation in SCLC due to the lack of ‘druggability’—intrinsic factors favoring the design of small chemical inhibitors. One notable success comes in a pharmacogenomic screening of *RB1*-mutant cancer cell lines that identified Aurora A, whose inhibition exhibited cell death selectively in *RB1*-mutant SCLC cells [66]. A CRISPR/Cas9-based screening identified *Aurora B* whose loss appeared to be synthetic lethal with *RB1* loss in SCLC [67]. The latter study speculated that inhibition of Aurora B might affect mitotic fidelity in *RB1*-deficient cells to a lethal level. Another study using an RNAi library in *RB1*-knockout lung cancer cells identified Aurora A and reasoned its mechanism by E2F-mediated upregulation of microtubule destabilizer Stathmin/OP18. Aurora A inhibition activated Stathmin through reduction of its phosphorylation, inducing mitotic cell death in *RB1*-knockout lung cancer cells [68].

Aurora kinases maintain mitotic fidelity by driving centrosome maturation, spindle assembly, centrosome bi-orientation and cytokinesis as one member of mitotic kinases. Many mitotic kinases are aberrantly expressed in *RB1*-deficient cells in an E2F-dependent manner and polo-like kinase 1 (PLK1) is one of them [69]. This provides one of the mechanisms whereby RB1 controls mitotic fidelity. Additionally, RB1 inactivation is linked to chromosomal instability (CIN) at least partially through enhanced replication fork stalling [69]. Thereby, in specific contexts, RB1 inactivation induces DNA damage response associated with increased activity of ATM, ATR, checkpoint kinase 1 (CHK1) and CHK2 and increased staining for γ-H2AX at the level that subsequently triggers cellular senescence [36,48,70,71,72]. A study using a pharmacological screening revealed that the synthetic CDK4/6 inhibitor strongly antagonizes the toxicity induced by a PLK1 inhibitor and a CHK1 inhibitor in triple negative breast cancer (TNBC) cells [73]. Conversely, the loss of *RB1* highly sensitized TNBC cells to these reagents. Actually, PLK1 and CHK1 were upregulated in *RB1*-deficient TNBCs. In the presence of PLK1 inhibitor, *RB1*-proficient cells arrest in G2/M, and thus escaped from cell death, but *RB1*-deficient cells maintain DNA replication and progress to higher aneuploidy and would hence undergo cell death when treated with PLK1 inhibitor. In the presence of CHK1 inhibitor, *RB1*-deficient cells exhibited several symptoms of S phase collapse that were absent in *RB1*-proficient cells [73]. These findings suggest that the unstable mitotic fidelity or dampen DNA damage response due to the inactivation of RB1, would expose tumor cells to novel vulnerability to agents that target these accumulated abnormalities to catastrophically trigger cell death, instead of resulting in CIN.

SCLC often harbor mutations in *RB1* and *Trp53,* but rare mutation in *BRCA1/2* nor other genes carrying homologous recombination functions. However, single usage of PARP inhibitors or combination therapy with another drug have been suggested to be effective in the treatment of SCLC [74]. Furthermore, in prostate cancers, simultaneous loss of *BRCA2* and *RB1* sensitizes tumor cells to PARP inhibitor [75]. In advanced prostate cancer cases, the co-deletion of *RB1* and *BRCA2* is frequently found. If even the lack of *RB1* alone increases the sensitivity to PARP inhibitors is not clarified yet.

There are indeed many observations from basic and clinical studies indicating that loss of RB1 function tend to associate with favorable response to chemotherapy [76]. More efficient and prolonged response to chemotherapy was reported in *RB1*-deficient non-small cell lung cancers (NSCLC) compared to in their *RB1*-proficient counterparts. This led the authors of these studies to propose that *RB1* deficiency has resulted in the bypassing of certain checkpoints [77]. Similarly, breast cancer cases with disrupted RB pathway tend to exhibit favorable response to neoadjuvant chemotherapy [78,79]. In TNBC, *RB1* deficiency may better benefit from gamma irradiation and chemotherapy using doxorubicin and methotrexate rather than *RB1*-proficient [80]. Similarly, the loss of RB1 function together with K-Ras mutation predicts favorable response to platinum-based chemotherapy in a particular type of pancreatic neoplasia [81]. More recently, *RB1* loss was identified as one of predictive markers of better response to a topoisomerase inhibitor irinotecan [82]. Therefore, whilst the loss of RB1 function is recognized as a hallmark of an overall poor prognosis for cancer patients, it may paradoxically provide clinically important vulnerabilities that could be targeted by genotoxic agents.

The acceleration of cell cycle progression following RB1 inactivation may provide further exploitable vulnerabilities in cancer therapy. Cell division cycle 25 proteins (CDC25s) are dual phosphatases, of which CDC25A and CDC25B are prominent in the frequency of their overexpression in cancers. Such a status in TNBC predicts poor prognosis. One of their most important substrates is cyclin dependent kinase 1 (CDK1), which cooperates with cyclin B to promote mitotic processes. Liu et al. screened a kinome/phosphatase inhibitor library for compounds cytotoxic to *RB1*, *PTEN* and *Trp53*-deficient TNBC cells, and identified a CDC25 inhibitor [83]. WEE1 is a nuclear kinase that phosphorylates CDK1 at the same site (Tyr^15^) as CDC25s [84]. The authors demonstrated that a WEE1 inhibitor synergizes with a CDC25 inhibitor. How inhibitors of proteins with such opposing functions synergize is explained by their distinct roles during cell cycle progression [85]. WEE1 phosphorylates CDK1 during S phase at Thr^14^ and Tyr^15^ enabling a separate phosphorylation at Thr^161^ by CDK-activating kinase (CAK) during G2 phase [86]. This additional phosphorylation at Thr^161^ allows CDC25s to fully activate CDK1 during M phase by executing dephosphorylation at Thr^14^ and Tyr^15^ [87]. In short, WEE1 and CDC25s cooperate to promote the M phase progression in a sequential manner [88]. Other researchers demonstrated that the treatment of TNBC cells with a CDC25 inhibitor collaterally activates a resistant mechanism that is supported by AKT activity [89]. Thus, an PI3K inhibitor would synergize with a CDC25 inhibitor. Although suppression of CDK1 activity was thought to be a key mechanism of the action of these inhibitors, CDK1 antagonist did not fully recapitulate it. This result indicates that there may be CDK1-indepdent functions of CDC25 and WEE1 in driving *RB1*-deficient cancers.

Prior to these reports, a study using *Drosophila melanogaster* system uncovered that inactivation of *rbf* (the fly orthologue of *RB1*) and *gig* (the fly orthologue of *tuberous sclerosis complex 2* (*TSC2*)) synergistically induces cell death in imaginal disc cells placed under stress conditions [90]. TSC2 together with TSC1 inhibits the function of a complex containing TORC1 and thereby suppresses tumor growth. Germline mutation in either of these genes causes an autosomal dominant tumor-prone syndrome. The phenotypes observed in *rbf-gig* double deficient fly cells appeared to depend on E2F transcription factors and the TOR (target of rapamycin) pathway. The synthetic lethal interaction between *RB1* and *TSC2* was also a case in multiple human prostate cancer cell lines. Inhibition of TSC2 in *RB1*-deficient cancer cells increased oxidative stress in a superoxide dismutase 2 (SOD2)-dependent manner, causing apoptosis [90]. The authors reasoned that this phenomenon was caused by the suppression of de novo lipid synthesis [90]. A follow-up study done by the same group indicated that *dtsc1* (the fly orthologue of TSC1) also exhibited synthetic lethality when inactivated together with *rbf*, which is associated with deregulated G1/S control, increased DNA damage and energy stress [91]. These findings suggest that the mTOR pathway may provide a vulnerable point in the treatment of *RB1*-deficient cancers in human.

An ‘ORFeome’ RNAi screen in *Caenorhabditis elegans* identified *zfp-2*, which encodes a zinc finger transcription factor, as a gene in a synthetic lethal relationship with *lin-35* (the *C. elegans* orthologue of *RB1*). These genes appeared to cooperate in somatic gonadal development [92].

A study made use of multiple publicly available large-scale shRNA and siRNA screening data bases to explore genes in the synthetic lethal relationship with *RB1* in TNBCs and newly identified NUP88 and NUP214 nuclear pore components associated with the MAD2 spindle check protein. It further explored a kinase and bromodomain containing transcription factor called transcription initiation factor TFIID subunit 1 (TAF1) and rediscovered SKP2 and SKP1 [93]. As above mentioned, MAD2 is overexpressed in *RB1*-deficient cells and critically contributes to CIN [69]. TAF1 has been demonstrated to directly bind to RB1 to carry out basal transcription [94].

The products of most of genes so far identified due to their synthetic lethal relationship with RB1 are pharmacologically targetable (Table 1). Although loss of RB1 function is generally associated with poor prognosis, it simultaneously gives rise to pharmacologically tractable vulnerabilities leading tumor cells to carry lethal levels of DNA damage, CIN, oxidative stress or E2F hyperactivation, causing cell death hopefully at a therapeutically satisfactory degree. As RB1 function is lost in a wide variety of cancer types at varied stages, synthetic lethality screening in *RB1*-deficient cells needs to be extended to more varies of cancers.

## 5. Targeting Loss of a Gene Neighboring *RB1*

The frequency that *RB1* gene loci undergo homozygous deletion in cancer is the fourth highest among tumor suppressor genes following *INK4A*, *PTEN* and *SMAD4* [95]. Genetic deletion of *SMAD4* that occurs in almost 30% of pancreatic ductal adenocarcinoma (PDAC) often associates simultaneous deletion of *malic enzyme 2* (*ME2*) gene, as these genes locate within 200 kb distance in the same chromosome. ME2 and its isozyme ME3 share oxidative decarboxylase function to convert malate to pyruvate and to regenerate NADPH. Depletion of ME3 in ME2-deficient pancreatic cancer cells induced lethal effects [96]. A similar collateral vulnerability has been found in the glycolytic gene *enolase 1* (*ENO1*) and *ENO2*. *ENO1* locates in the 1p36 locus which is frequently deleted in glioblastoma [97]. These studies demonstrated that targeting genes involved in genomic collapse in the chromosomal region surrounding tumor suppressor genes or frequently collapsed genome regions may generate therapeutic benefits through collateral lethality.

Kohno et al. attempted a survey of metabolic genes located near to tumor suppressor genes that are frequently deleted in cancers. They then examined the frequency of simultaneous loss of such genes upon tumor suppressor gene loss in individual cancer type. This effort discovered *succinyl lyase A2* (*SUCLA2*) gene that locates around 302 kb away from *RB1* tumor suppressor gene in human chromosome 13q14.2. The germ line mutation or deletion in *SUCLA2* gene is causative of mitochondrial encephalomyopathy and methylmalonic aciduria [98]. There is a case report of a patient who had undergone a genomic deletion involving both *SUCLA2* and *RB1* gene. The patient developed encephalomyopathy and retinoblastoma simultaneously [99]. The simultaneous loss of *RB1* and *SUCLA2* was detected in various cancer types with varied prevalence. Surprisingly, particularly in prostate cancers, *SUCLA2* gene deletion was found to take place upon *RB1* loss with near to 100% prevalence [100].

SUCLA2 contributes to a complex of enzymes that converts succinyl-CoA to succinate in the TCA cycle and generates GTP from GDP. Indeed, depletion of SUCLA2 in prostate cancer cells resulted in lower mitochondrial/respiratory activity and enhanced glycolysis. These metabolic phenotypes continuously sustained as long as SUCLA2 was downregulated and thus were likely to be targeted aiming for treatment. Kohno et al. sought the ways to make use of this finding. First, they attempted to additionally inhibit SUCLG2 isozyme in the presence of *RB1-SUCLA2* deletion, as it might completely disrupt the conversion from succinyl-CoA to succinate. However, it did not kill such cells with satisfactory efficacy [100].

Kohno et al. next screened a number of chemical libraries to find compounds that selectively kill *RB1-SUCLA2*-deficeint prostate cancer cells [100]. This attempt hit several compounds including 2-isopropyl-5-methylbenzo-1,4-quinone (thymoquinone), a natural compound which is known as one of major bioactive ingredients of *Nigella Sativa* [101] (Figure 1). Although its anticancer and antioxidative functions were reported, the molecular target has been unidentified. In vivo treatment of immunodeficient mice xenografted with *RB1-SUCLA2*-deficeint prostate cancer cells with thymoquinone resulted in a significantly selective suppression of tumorous growth via apoptosis [100]. These findings indicated that the loss of a gene neighboring *RB1* can be pharmacologically targeted in cancer therapy. Homozygous deletion of *RB1* is estimated to occur in more than 30% of advanced prostate cancer cases [11]. Heterozygous deletion of *RB1* is also frequent in advanced prostate cancers, which is naturally associated with heterozygous loss of *SUCLA2*. Thymoquinone appeared to be effective to *SUCLA2*-heterozyous prostate cancers as well [100]. Interestingly, the Aurora A inhibitor that targets *RB1* loss in a synthetic lethal manner synergized with thymoquinone in prostate cancer cells doubly deficient of *RB1* and *SUCLA2* [100]. Furthermore, more varieties of cancers (bladder, ovarian, uterine, liver, cervical, esophageal, breast cancers, sarcoma, diffuse large B-cell lymphoma, etc.) are estimated to carry *RB1-SUCLA2* deletion with up to 5% prevalence [100]. The development of drugs with stronger thymoquinone-like activity and identification of target molecule are awaited. Additionally, the discovery of biomarkers that help quick diagnosis of *SUCLA2* deficiency in tumor tissue would be forthcoming [100].

In addition to SUCLA2, there are some more genes neighboring *RB1* locating even closer to *RB1*. Lee et al. demonstrated that among such genes, loss of *integral membrane protein 2B* (*ITM2B*), *chromosomal condensation 1-llike* (*CHC1L*) and *purinergic receptor P2Y G protein-coupled protein 5* (*P2RY5*) function may contribute to enhance tumor promotion in *RB1*-deficient bladder cancer [102]. In a sharp contrast, loss of SUCLA2 would endow a certain disadvantage from the views of metabolism and pharmacological vulnerability. Probably, this explains why *RB1-SUCLA2* co-deletion is not quite prevalent in cancers other than prostate cancer [100].

Dyson’s group carried out shRNA screenings to identify genes assisting RB1-induced cellular senescence and discovered large tumor suppressor 2 (LATS2) [103]. Interestingly, *LATS2* locus are frequently deleted simultaneously with *RB1* locus. LATS2 is primarily a component of Hippo pathway, but appeared to phosphorylate dual-specificity tyrosine-phosphorylation-regulated kinase 1A (DYRK1A), which then phosphorylates LIN52 a subunit of DREAM (DP-1, RB1, E2Fs and MuvB) complex. Therefore, loss of LATS2 impairs functions of DREAM complex, cooperating with *RB1* loss to promote tumor development [104]. In this scenario, in sharp contrast to SUCLA2, loss of *LATS2* generates certain growth advantages to *RB1*-deficient cancer. It is of interest to address if pharmacological enhancement of LATS2 function in *RB1*-deficient cancer is feasible. As described above, *BRCA2* gene is also often co-deleted with RB1 [74]. These tumors are good therapeutic targets of PARP inhibitors.

## 6. Conclusions

Inactivation of RB1 functions and the RB pathway is a major hallmark of cancer. Irreversible dysfunction of RB1 often predicts poor prognosis in many types of cancer. However, the *RB1*-defecient status in cancers may turn out to be a disguised blessing for cancer patients as it generates a variety of therapeutic benefits through the upregulation of RB1 targets, the presence of genes which are in a synthetic lethal relationship with *RB1*, and collateral synthetic lethality due to *SUCLA2* loss (Figure 2).

## Figures and Tables

**Figure 1 cancers-13-03737-f001:**
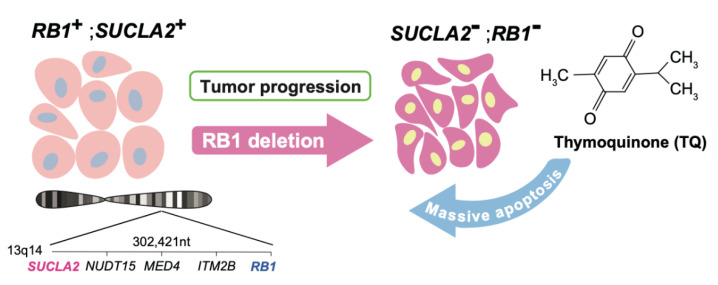
Thymoquinone targets *SUCLA2* loss that collaterally takes place with *RB1* homozygous or heterozygous deletion.

**Figure 2 cancers-13-03737-f002:**
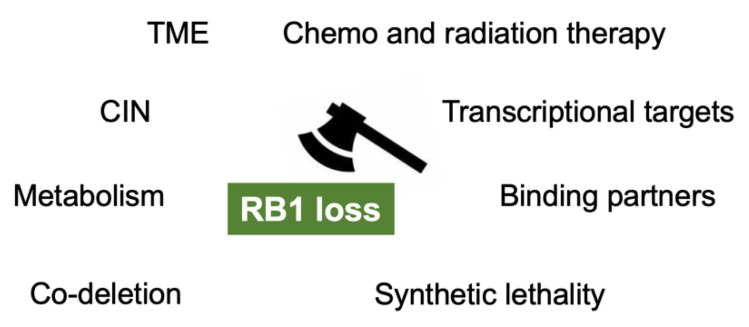
The mechanisms which enable us to target *RB1* loss in cancers. TME: tumor microenvironment; CIN: chromosomal instability.

**Table 1 cancers-13-03737-t001:** Genes whose inactivation exerts synthetic lethality in the presence of *RB1* deficiency, and representative inhibitors to the gene products and phase of clinical trials. ND: no data. The information was obtained by searching online accessible catalogues provided by Selleck, Santa Cruz and Abcam.

Gene	Inhibitor	Phase of Clinical Trial
Aurora A, B	Alisertib	III
	Tozasertib	II
	Barasertib	I
	MNL8054	I
	Danusertib	II
	AT9283	II
	KW-2449	I
	SNS-341	I
	ENMD-2076	II
	BI-847325	I
PLK1	Volasertib	III
	Rigosertib	III
	BI2536	II
CHK1	AZD7762	I
	MK-8779	II
	PF-477736	I
CDC25A, B	K-252a	ND
	NSC663284	ND
	NSC95397	ND
WEE1	Adabosertib	II
TSC1, 2	ND	ND
SKP2	SKPin C1	ND
TAF1	CeMMEC-1, 13	ND
	BAY299	ND
NUP88	ND	ND
NUP214	ND	ND
PARP	Olaparib	FDA-approved
	Veliparib	III
	Rucapatib	III
	Iniparib	III
SYK	Fostamatinib	III
	R406	I

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
