# Peer review of "Targeting RB1 Loss in Cancers"

_cancers, 2021, doi:10.3390/cancers13153737_

Round 1

Reviewer 1 Report

RB1 is a well-characterised tumor suppressor protein that is often impaired in solid and metastatic tumors. Here in this review, Linn and colleagues have summarised the current finding of RB1 loss in various cancer types and how this can be exploited for therapeutic targeting. The overall manuscript is well-written and touches on the current treatment approaches including on-going clinical trials. Several areas of the manuscript can be further strengthened. In the introduction section, the authors can potentially include a table of the top 10 or 20 malignant tumors with RB1 loss before jumping into the different cancer types and discuss how it affects cancer progression (Cancer Discovery. 2017;7(8):818-831). It will be favourable to include a short discussion of high-grade glioma, head and neck cancer and sarcoma as these cancers are also known to have a significant patient population with RB1 loss (See TCGA for more information). As for section 2, RB1 loss can predominantly dysregulate mitotic progression, DNA damage response, DNA replication and cell-cycle. While the authors were able to discuss the role of transcriptional factors and mitotic events as a result of RB1 loss, a more balanced discussion to include DNA damage response, DNA replication and cell-cycle is justified (See Knudsen et al 2020, Communications biology 3, article 158 and Knudsen et al 2019, Trends Cancer 5(5), 308–324). In the last paragraph of section 3, it would be appropriate to provide context where the loss of RB1 can promote undifferentiated properties of cancer cells (ie: Will the loss of RB1 promotes certain stemness genes or prevent cell differentiation from occurring? Is it cancer type specific? Youn et al 2013, Nature immunology 14, 211-220). Additionally, there should be discussion of RB1 loss in immunotherapy response and also RB1 loss in acquired resistance in cancers since this review has a specific focus on therapeutic targeting.

Author Response

Response to Reviewer #1: 

RB1 is a well-characterized tumor suppressor protein that is often impaired in solid and metastatic tumors. Here in this review, Linn and colleagues have summarized the current finding of RB1 loss in various cancer types and how this can be exploited for therapeutic targeting. The overall manuscript is well-written and touches on the current treatment approaches including on-going clinical trials.

  1. In the introduction section, the authors can potentially include a table of the top 10 or 20 malignant tumors with RB1 loss before jumping into the different cancer types and discuss how it affects cancer progression (Cancer Discovery. 2017;7(8):818-831).

 We are able to provide a table that covers 10 to 20 types of cancer with RB1 mutation or loss. However, the same way of demonstration has been already performed by Viatour and Sage a while ago [REF #7]. To avoid overlap with their review article, we would not like to abide by the suggestion kindly made by this reviewer. Instead, we described 9 types of corresponding malignancy in the text. Additionally, the reviewer kindly advised us to refer to Cancer Discovery. 2017;7(8):818-831. However, this article appeared to discuss on how cancer genomic and clinical information should be shared by multiple institutions, and hence provides few information on RB1 status in cancers. Accordingly, we did not refer to this article in our manuscript.

  1. Several areas of the manuscript can be further strengthened. It will be favorable to include a short discussion of high-grade glioma, head and neck cancer and sarcoma as these cancers are also known to have a significant patient population with RB1 loss (See TCGA for more information).

 We highly appreciate this nice suggestion and included these tumors. Additionally, we gave a short discussion on sarcoma.

3. As for section 2, RB1 loss can predominantly dysregulate mitotic progression, DNA damage response, DNA replication and cell-cycle. While the authors were able to discuss the role of transcriptional factors and mitotic events as a result of RB1 loss, a more balanced discussion to include DNA damage response, DNA replication and cell-cycle is justified (See Knudsen et al 2020, Communications biology 3, article 158 and Knudsen et al 2019, Trends Cancer 5(5), 308–324).

 In the previous version, due to given word number limitation of manuscript, we referred to only pharmacologically targetable transcription factors involved in mitotic regulation and therefore skipped discussion on DNA damage response, DNA replication and cell-cycle. In the revised manuscript, we provided additional discussion on these issues at minimum requirement. We found that the latter review article written by Dr. Knudsen is highly helpful, thus cited it as well as other appropriate articles. Thank you so much for your insightful comments.

  1. In the last paragraph of section 3, it would be appropriate to provide context where the loss of RB1 can promote undifferentiated properties of cancer cells (ie: Will the loss of RB1 promotes certain stemness genes or prevent cell differentiation from occurring? Is it cancer type specific? Youn et al 2013, Nature immunology 14, 211-220).

 According to this suggestion, we provided another paragraph after this for extensive discussion. We discussed on the change in undifferentiated status flowing RB1 loss in iPSCs, breast cancer and prostate cancer. As we still have only small number of evidence, we avoided to further discuss on whether it is cancer type specific or not. Hence, we did not cite the paper that this reviewer suggested.

  1. Additionally, there should be discussion of RB1 loss in immunotherapy response and also RB1 loss in acquired resistance in cancers since this review has a specific focus on therapeutic targeting.

 This is a very nice suggestion. We discussed on RB1 loss in immunotherapy response and acquired resistance in cancer.

Reviewer 2 Report

This review by Linn et al entitled “Targeting RB1 loss in cancers” provides an appropriate, yet brief, overview of the field as it pertains to therapeutic targets in RB1 deficient cancers. However, the authors frequently cite review articles when they should instead cite and give credit to the authors responsible for the original research. The review would benefit from the inclusion of additional discussion and citations beyond the single examples provided for several subsections. The review would also benefit from a more comprehensive model figure.

Specific Comments:

  1. the authors have overlooked synthetic lethal interactions between RB loss and SYK inactivation published by the Dyer lab (Zhang et al., 2012) and between RB and HELLS published by the Dyer and Benevente labs (Benevante et al., 2014; Zocchi et al., 2020) and between RB loss and LATS2 (Tschop et al., 2011; Tschop and Dyson, 2011). These relationships would be appropriately discussed in section 2. “Targeting gene products upregulated following RB1 inactivation” and/or section 5. “Targeting loss of a gene neighboring RB1”

  1. in the paragraph beginning on Line 91 the authors introduce that RB1 inactivation induces CIN, but cite only one of several distinct mechanistic papers that describe how RB1 may regulate mitotic progression. Work from Amato et al., 2009; Coschi et al., 2010; and Manning et al., 2010 should also be discussed and cited in this context.

  1. Sanidas et al., 2019 (or prior reviews from the Dyson Lab and others) should be cited to substantiate statements regarding RB1 binding partners

  1. line 58 indicates stem cell like behavior following RB1 inactivation is specific to breast cancers, though the citations provided investigate several different cancer contexts. The citations and/or text should be revised to appropriately reflect what ios described in these research articles.

  1. The citation is missing from the statement that the blockade of IL-6 by neutralizing antibody or inhibition of CCR2 function suppressed breast cancer progression (lines 58-60)

  1. The citation for Rouaud et al., 2018 should be included in the discussion of E2F inhibition

  1. Citations are missing from statements in line 148

  1. Lines 140-155 require additional detail to add meaningful content to this review- which transcription factors? How are they thought to collaborate with RB1? What are the consequences of dual loss/inhibition? Citations of the primary research articles are needed here (not just a review article)

  1. Line 185, line 197, and others- the primary research articles should be cited here, not a review article!

  1. Line 188 should include citation of Velez-Cruz et al., 2016 and Marshall et al., 2019

  1. Citation #59 appears to be a manuscript about whale sharks, and is not the appropriate citation for sensitivity of RB deficient cells to doxorubicin (line 208-209)

  1. Numerous citations to primary research articles are missing to support statements on WEE1 and CDC25 (lines223-237)

  1. Its unclear why section 5 “Targeting loss of a gene neighboring RB1” is restricted to discussion of SUCLA2? The review woudlbe more comprehensive if the authors also considered other co-deleted genes that may influence drug sensitivity of RB1 deficient cancers (see Tschop et al., 2011 regarding LATS2, etc) and Lee et al., 2007 regarding ITM2B, CHC1L, and P2RY5)

  1. line 288: Kono et al study is introduced by the corresponding in text citation is not present. This same citation should be referenced again on line 308.

  1. Line 309: its unclear who “they” are? Khono et al.,? or Khader & Eckl, 2014 (as the citation seems to suggest)? If not Khader & Eckl, please provide the appropriate citations.

  1. Discussion of synthetic lethality of RB1 loss and PARP inhibition would be appropriate to discuss in section 4 of this review (Farago et al., 2019 and Chakraborty et al., 2020)

  1. It would be useful to expand table 1 to include additional hypothetical drug targets that have been discussed in this review (ie Mad2). Also missing from this table are inhibitors of PARP and SYK.

  1. While the cartoon in Figure 1 adequately demonstrates a single synthetic relationship, a figure more broadly representing the RB1-dependent pathways (overexpressed genes, CIN, replication, metabolism) and the corresponding clinical targets (as discussed in this review) would be more appropriate and informative

    19. citation #27 does not appear to be relevant to the statements on       line 105 (or anything else in this review). Additional/alternative citations are needed to support these statements

Grammar and formatting comments

  1. It is inaccurate to consider 20-year old articles to be ‘recent’ (line 106) and the authors should consider alternate wording

  1. There are numerous grammatical and spelling errors throughout that should be fixed and the manuscript would benefit from careful proofreading. Some examples include (from the intro alone):

  • Line 34: Together with the presence of another cancer driving mutations such as that of . should be Together with the presence of other cancer…
  • Line 38 should read: suggesting a contribution to their initiation
  • Line 41-42: RB1 reconstitution in RB1-deficient osteosarcoma-derived cells does not completely attenuates
  • Line 43-44 should read: These findings indicate that RB1 may…

  1. There are several instances where BOTH the numerical citation and author name, year format are used for a single reference (Example on lines 176 and 185)

    4. Citation #64 has two years of publication listed- 2014 and 2013.

Author Response

Response to Reviewer #2: 

This review by Linn et al entitled “Targeting RB1 loss in cancers” provides an appropriate, yet brief, overview of the field as it pertains to therapeutic targets in RB1 deficient cancers. However, the authors frequently cite review articles when they should instead cite and give credit to the authors responsible for the original research. The review would benefit from the inclusion of additional discussion and citations beyond the single examples provided for several subsections. The review would also benefit from a more comprehensive model figure.

Specific Comments:

  1. the authors have overlooked synthetic lethal interactions between RB loss and SYK inactivation published by the Dyer lab (Zhang et al., 2012) and between RB and HELLS published by the Dyer and Benevente labs (Benevante et al., 2014; Zocchi et al., 2020) and between RB loss and LATS2 (Tschop et al., 2011; Tschop and Dyson, 2011). These relationships would be appropriately discussed in section 2. “Targeting gene products upregulated following RB1 inactivation” and/or section 5. “Targeting loss of a gene neighboring RB1”

 We really appreciate this great suggestion. We introduced all of these papers in the manuscript. We indeed overlooked SYK and HELLS. We respect the wisdom of this reviewer. Not like SUCLA2, LATS2 deletion most probably promotes tumor growth when combined with RB1 mutation, so, LATS2 deletion may not be a good therapeutic target. However, we introduced this at the end of the section, although LATS2 is not a neighbor of RB1.

  1. In the paragraph beginning on Line 91 the authors introduce that RB1 inactivation induces CIN, but cite only one of several distinct mechanistic papers that describe how RB1 may regulate mitotic progression. Work from Amato et al., 2009; Coschi et al., 2010; and Manning et al., 2010 should also be discussed and cited in this context.

  We appreciate this suggestion. The purpose of this paragraph is to shed light on ‘targetable molecules’ that are involved in RB1-mediated mitotic regulation, not to comprehensively introduce numerous molecules that may not be targetable. Amato et al., 2009 probably corresponds to BMC Cell Biol. 2009;10:79. This indicated Aurora A and polo-like kinase 1 (PLK1) are upregulated following RB1 depletion thus can be therapeutic target. We are glad to inform this reviewer that we cited this paper together with a similar report by Sage and Straight (2010). Coschi et al., 2010 from Dick lab described mitotic defects induced by RB1 loss in p53-/- background but never identified molecules that are involved in the phenomenon and pharmacologically targetable. Thus, we did not cite this paper. Manning et al., 2010 (Gen Dev) provides lots of information on CAPD3/condensin II, which we cited.

  1. Sanidas et al., 2019 (or prior reviews from the Dyson Lab and others) should be cited to substantiate statements regarding RB1 binding partners.

 ïƒ  We cited this article, which made our review much more informative. Thank you!

  1. line 58 indicates stem cell like behavior following RB1 inactivation is specific to breast cancers, though the citations provided investigate several different cancer contexts. The citations and/or text should be revised to appropriately reflect what ios described in these research articles.

 We did not mention that stem cell like behavior following RB1 inactivation is specific to breast cancers. To avoid confusion, we added comments on other cancers.

  1. The citation is missing from the statement that the blockade of IL-6 by neutralizing antibody or inhibition of CCR2 function suppressed breast cancer progression (lines 58-60)

 ïƒ We clarified citation.

  1. The citation for Rouaud et al., 2018 should be included in the discussion of E2F inhibition

  ïƒ  We cited this article, which made our review much more informative. Thank you!

  1. Citations are missing from statements in line 148

 ïƒ  We cited an appropriate information. Thank you!

  1. Lines 140-155 require additional detail to add meaningful content to this review- which transcription factors? How are they thought to collaborate with RB1? What are the consequences of dual loss/inhibition? Citations of the primary research articles are needed here (not just a review article)

 We defined transcription factors which I meant here; MYOD, C/EBP, GR, GATA-1, PU-1, CBFA-1, PDX1, RUNX2, and NF-IL although we could not cover all. As the interaction of RB1 and these transcription factors has been repeatedly described by numerous previous review articles and also we mentioned targeting these factors could be therapeutically not feasible, i.e., this part is not the main focus of this review article, we decided to cite only two representative review articles.

  1. Line 185, line 197, and others- the primary research articles should be cited here, not a review article!

 ïƒ We agree with this suggestion. For line 185, we inserted an original research article as a reference. However, for lie 197 in the initially submitted version, we already cited an original research article and explained whole story totally according to this paper. Thus, we did not rewrite this particular part.

  1. Line 188 should include citation of Velez-Cruz et al., 2016 and Marshall et al., 2019

 We were pleased to additionally cite Velez-Cruz et al., 2016 and Marshall et al., 2019.

  1. Citation #59 appears to be a manuscript about whale sharks, and is not the appropriate citation for sensitivity of RB deficient cells to doxorubicin (line 208-209)

We have completely mistaken the citation. We changed this to an appropriate one.

  1. Numerous citations to primary research articles are missing to support statements on WEE1 and CDC25 (lines223-237)

 ïƒ  We are sorry that we depended on too small number of references for this part. We increased the number of references in the manuscript as much as possible.

  1. It’s unclear why section 5 “Targeting loss of a gene neighboring RB1” is restricted to discussion of SUCLA2? The review would be more comprehensive if the authors also considered other co-deleted genes that may influence drug sensitivity of RB1 deficient cancers (see Tschop et al., 2011 regarding LATS2, etc) and Lee et al., 2007 regarding ITM2B, CHC1L, and P2RY5)

 This is the really great suggestion. We of course thought about other genes neighboring RB1 but could not identify related publications. LATS2 is often co-deleted with RB1, but this locates in a different chromosome and itself seems to be a tumor suppressor. However, the ITM2B, CHC1L and P2RY5 story is really informative in the section 5. In contrast to SUCLA2 which loss gives rise to growth disadvantage and pharmacological vulnerability, the collateral lack of these genes seems to be tumor promoting. Thus, unlike with SUCLA2, lack of these genes might not be good therapeutic target. Anyway, thank you so much for this excellent input. We hope this reviewer appreciate our new discussion.

  1. Line 288: Kohno et al study is introduced by the corresponding in text citation is not present. This same citation should be referenced again on line 308.

 We cited this again in the appropriate parts.

  1. Line 309: its unclear who “they” are? Khono et al.,? or Khader & Eckl, 2014 (as the citation seems to suggest)? If not Khader & Eckl, please provide the appropriate citations.

 ïƒ ’They’ means Kohno et al.,. We clarified this.

  1. Discussion of synthetic lethality of RB1 loss and PARP inhibition would be appropriate to discuss in section 4 of this review (Farago et al., 2019 and Chakraborty et al., 2020)

 ïƒ Thank you very much for reminding us of these important works. We introduced all of these papers.

  1. It would be useful to expand table 1 to include additional hypothetical drug targets that have been discussed in this review (ie Mad2). Also missing from this table are inhibitors of PARP and SYK.

 ïƒ We are pleased to include information on PARP and SYK in the new table 1.

  1. While the cartoon in Figure 1 adequately demonstrates a single synthetic relationship, a figure more broadly representing the RB1-dependent pathways (overexpressed genes, CIN, replication, metabolism) and the corresponding clinical targets (as discussed in this review) would be more appropriate and informative.

We newly generated Figure 2 according to this kind advice.

  1. citation #27 does not appear to be relevant to the statements on line 105 (or anything else in this review). Additional/alternative citations are needed to support these statements

We are very sorry that we mistakenly cited this article. We changed this to a correct one.

Grammar and formatting comments

  1. It is inaccurate to consider 20-year old articles to be ‘recent’ (line 106) and the authors should consider alternate wording

 This is apparently a mistake. We changed this to ‘while ago’.

  1. There are numerous grammatical and spelling errors throughout that should be fixed and the manuscript would benefit from careful proofreading. Some examples include (from the intro alone):

 ïƒ  We asked a professional English editor for improving our manuscript. Thank you so much picking up examples.

  • Line 34: Together with the presence of another cancer driving mutations such as that of should be Together with the presence of othercancer…
  • Line 38 should read: suggesting a contribution totheir initiation
  • Line 41-42: RB1 reconstitution in RB1-deficient osteosarcoma-derived cells does not completely attenuates
  • Line 43-44 should read: These findings indicatethat RB1 may…
  1. There are several instances where BOTH the numerical citation and author name, year format are used for a single reference (Example on lines 176 and 185)

 We are very sorry that we forgot to erase these.

  1. Citation #64 has two years of publication listed- 2014 and 2013.

 Most probably, this reviewer is saying about the page number. We confirmed this information is correct.

Round 2

Reviewer 1 Report

Authors have revised manuscript according to reviewer’s comments. The manuscript reads well and includes a more comprehensive discussion on the overall view of RB1 loss in cancers. There are no major concerns and should warrant publication.